

# Density structure and isostasy of the lithosphere in Egypt and their relation to seismicity

Mikhail K. Kaban[1], Sami El Khrepy[2,3], Nassir Al-Arifi[2]

[1]GFZ German Research Centre for Geosciences, Telegrafenberg A 20, D-14473 Potsdam, Germany
[2]King Saud University, Riyadh, Saudi Arabia, P.O. Box 2455, Riyadh 11451, Saudi Arabia
[3]National Research Institute of Astronomy and Geophysics, NRIAG, 11421, Helwan, Egypt

*Correspondence to*: Mikhail K. Kaban (kaban@gfz-potsdam.de) and Sami El Khrepy (selkhrepy@ksu.edu.sa)

**Abstract.** A joint analysis of the new satellite-terrestrial gravity field model with the recent data on the crustal structure and seismic tomography model was conducted to create an integrative model of the crust and upper mantle; and to investigate the
relation of the density structure and the isostatic state of the lithosphere to the seismicity of Egypt. We identified the distinct fragmentation of the lithosphere of Egypt into several blocks. This division is closely related to the seismicity patterns in this region. The relatively dense and strong lithosphere in the Nile Delta limits the seismic activity within this area, while earthquakes are mainly associated with the boundaries of this block. In the same way, the relatively strong lithosphere in the Suez Isthmus and northern Mediterranean prevents the Gulf of Suez from opening further. The central part of Egypt is
generally characterized by an increased density of the mantle, which extends to the Mediterranean at a depth of 100 km. This anomaly deepens southward to Gilf El Kebir and eastward to the Eastern Desert. The average density of the crystalline crust is generally reduced in this zone, indicating the increased thickness of the upper crust. The low-density anomaly under the northern Red Sea is limited to 100–125 km, confirming the passive origin of the extension. Most of the earthquakes occur in the crust and uppermost mantle in this structure due to the hot and weak upper mantle underneath. Furthermore, an
asymmetric lithosphere structure is observed across the Northern Red Sea. The isostatic anomalies show the fragmentation of the crust of Sinai with the high-density central block. Strong variations of the isostatic anomalies are correlated with the high level of seismicity around Sinai. This tendency is also evident in the North Red Sea, east of the Nile Valley, and in parts of the Western Desert.

## 1 Introduction

A thorough understanding of the solid Earth system is an essential step towards deciphering the link between the dynamic processes in the Earth system and near-surface processes. In particular, the density heterogeneity of the lithosphere and upper mantle largely controls tectonic processes, which in turn produce strong density perturbations in the upper crust. Therefore, the knowledge of density variations is essential to understand the structure and dynamics of the lithosphere. Up to now, seismological methods have been a key to unravelling the structure of the crust and upper mantle and provide an
increasingly detailed image of the interior of the Earth. However, they cannot provide a complete image of the structure of





the crust and upper mantle. For example, seismic velocities in the upper mantle are more sensitive to temperature than to compositional variations (e.g., Tesauro et al., 2014); therefore, tomography images primarily reflect temperature variations. A clear example is represented by high-density eclogitic rocks, which are characterized by seismic velocities that are close to normal upper mantle conditions and therefore are almost invisible in seismic models (Krystopowicz and Currie 2013). It is

also important that different seismological methods provide different estimates of various parameters. On the other hand, the gravity field of the Earth, which directly images density variations, cannot be used separately to model density heterogeneity. The inverse gravity problem is essentially ill-posed and its solution depends completely on initial model assumptions. Therefore, the identification of density variations in the crust and upper mantle is challenging and this problem cannot be solved by a particular geophysical method alone. Recent efforts aimed at integrating multiple geophysical and petrological

datasets in a common interpretation framework (e.g., Fullea et al., 2009; Gradmann et al., 2013; Kaban et al., 2001; Kaban et al., 2014a).

Many seismic studies have been performed in Egypt; however, most of them are related to the northern and eastern parts of the country, along the Red Sea and Mediterranean. With respect to recent results, we refer to Abdelwahed et al. (2013) showing the Conrad and Moho discontinuities of Eastern Egypt and the Red Sea. Corchete et al. (2017) determined the

crustal and uppermost mantle structure in northeastern Egypt based on Rayleigh wave analysis. Hosny and Nyblade (2016) determined the vertical $V_s$ sections of the crust and Moho depth for 26 stations in Egypt; however, only few of them are located in the central and south-eastern parts. Most of the previous seismic determinations of the crustal structure were summarized in Stolk et al. (2013) in the crustal model of Asia, which also includes Northeast Africa. These studies show that the coverage of the territory of Egypt based on seismic methods is very heterogeneous. While the areas adjoining the Red

Sea and Mediterranean are well studied (El Khrepy et al., 2015 and 2016, Hosny and Nyblade (2016), Mohamed et al., (2014), the data for the central, western, and southern parts are sparse. Several important questions remain unresolved. The main question relates to the structural division of the lithosphere and to what extent the surface tectonic units are related to the deep heterogeneity of the crust and upper mantle.

Recent satellite gravity missions (mainly GRACE and GOCE) provided the possibility to produce new generation gravity

models based on the combination of satellite and terrestrial data (e.g., Förste et al., 2014). These models stimulated new studies of the crustal structure and particularly the determination of the Moho boundary in Egypt and surrounding areas (e.g., Azab et al., 2015; Cowie and Kusznir, 2012; Prutkin and Saleh, 2009; Salem et al., 2013; Sobh et al., 2016). The results obtained in these studies are very controversial. As already mentioned, the solution of the ill-posed inverse gravity problem highly depends on initial assumptions. The Earth's gravity field is induced by the density heterogeneity of the entire planet;

therefore, its inversion with respect to 1–2 parameters often provides biased results. Only an interpretation integrating all available geophysical, geological, and mineral physics data might help to overcome this internal weakness of the gravity approach. Kaban et al. (2016c) presented an integrative model based on a joint analysis of seismic and gravity data for the



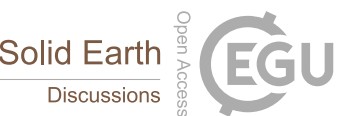

entirety of the Middle East, which partially covers the area of the present study. However, for the territory of Egypt this model is not defined in that study due to the lack of data, in particularly on the crustal structure. Another problem is related to the evaluation of the isostatic state of the lithosphere, which is often related to seismicity (e.g., Assumpção and Sacek, 2013; Sobiesiak et al., 2007). Segev et al. (2006) published a comprehensive study on this topic for the Levant continental

margin and the southeastern Mediterranean area; however, this study only marginally touches Egypt.

One of the motivating objectives is to find an interpretation for the very low seismicity pattern in northern Egypt (ENSN earthquake Catalogues, 1997-2016) in relation to lithosphere structure in this region, which includes highly populated areas and intensive international trade ways. On the other hand, the high level of seismicity, the shallow depth of hypocenters in the northern Red Sea including the shear zones along its western coast should be also investigated with respect to the

structure of the crust and upper mantle and to the isostatic state of the lithosphere. The asymmetric seismicity pattern in the northern Red Sea is another subject for discussions. Furthermore, the termination of the Gulf of Suez rift without continuation to the Mediterranean Sea will be also discussed in relation to the density structure of the lithosphere.

In the present study, we use an integrative interpretation of gravity, seismic, geological, and mineral physics data for the investigation of the density structure of the crust and upper mantle in Egypt and its surroundings, and to evaluate the isostatic

state of the lithosphere and its relation to the seismicity in this region. Such kind of an integrative geophysical study is applied to the study area for the first time; this was made possible due to the availability of new data after establishing in 1997 the Egyptian National Seismic Network (ENSN), which provided the input for new receiver function and tomography models of the Egyptian lithosphere (e.g. El khrepy et al., 2015, 2016; Abdelwahed et al., 2013; Hosny and Nyblade 2016; and Mohamed et al., (2014)**.**

## 2 Tectonic settings and seismicity of Egypt

The lithosphere of Egypt formed in a very active tectonic frame. In the north, it is bounded by the continental collision zone in which the African Plate subducts under Eurasia with a velocity of approximately 6 mm/year (McClusky et al., 2000). The left-lateral strike-slip Dead Sea Transform continues to the Aqaba Fault Zone at the north-eastern boundary. On the eastern side, the Red Sea represents an active extension zone dividing Africa and Eurasia at a variable extension rate, which

increases from the north (~5.6 mm/year) to the south (14 mm/year; McClusky et al., 2003). This active environment produces high and continuous seismic activity in the region (Fig. 1). Sinai represents a sub-block of the lithosphere, which is bounded by the Aqaba and Suez fault zones. It is still under debate as to which principal continental plates, African or Eurasian, it can be attributed with respect to its deep structure and dynamics. West of the Red Sea, the Eastern Desert extends to the Nile Valley. In the south, it is a part of the Nubian Shield, while the northern and north-eastern parts are

covered by Eocene sediments. The Western Desert covers the majority of Egypt west of the Nile Valley. The Gilf Kebir Plateau is located in the southwestern corner of Egypt (Fig. 1).

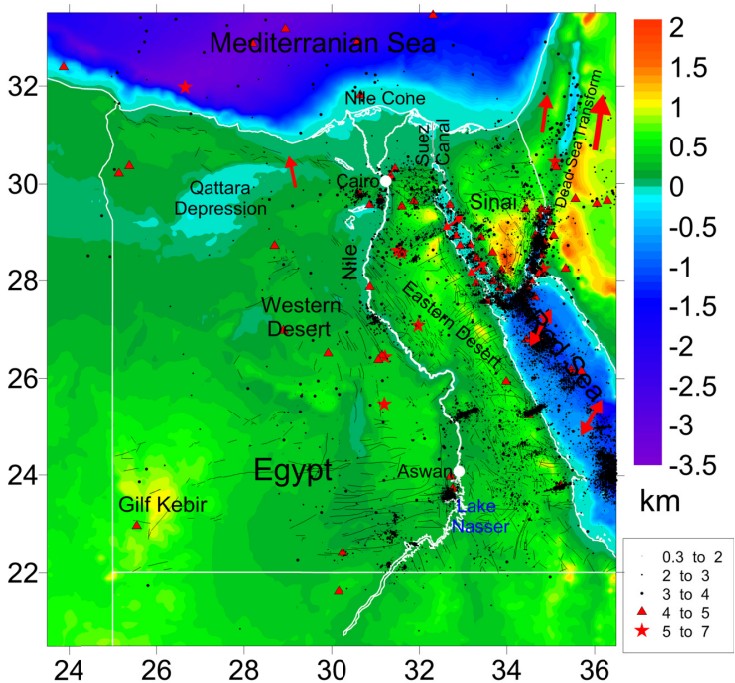

**Figure 1.** Topography of Egypt and surrounding area. Black dots, red triangles and stars show seismicity of Egypt (Egyptian National seismological Catalogue ENSN 1998-2011) seismicity. Black lines show faults (Egyptian Geologic Survey and Mining Authority, EGSMA, 1992). Red arrows demonstrates principal trend of the plate motions (Stern and Johnson, 2010).

Due to its location in the north-eastern African continent, seismic activity in Egypt is mainly controlled by regional stresses from active tectonic surroundings. The interaction of tectonic processes results in different levels of seismicity in Egypt, mostly limited to the crust and upper mantle (Fig 1). Clustered and scattered earthquake activity are well defined to the parallel shear zones along the western Coast of the Red Sea, the entrance and axial trough of the Gulf of Suez, the Cairo Suez Region, and the eastern part of Egypt. The most intensive zones of earthquake activity and earthquake swarms are located in the North Red Sea, the southern Sinai tip, and the two gulfs of Aqaba and Suez—especially at the intersection of the plate boundaries—while the western part of Egypt is seismically stable, with no remarkable activity (El Khrepy et al., 2015, 2016). The seismicity pattern is directed NW–SE in the Red Sea and Gulf of Suez, in accordance with the Red Sea rift





direction. The earthquakes in the Gulf of Aqaba tend NE in the direction of the Dead Sea Rift. The seismicity of the northern coast of Egypt is related to Eastern Mediterranean tectonics associated with the surrounding plate boundaries (Cyprian and Hellenic arcs, Anatolian Fault System) (Fig 1). The seismicity along the Nile River occurs in scattered cluster patterns; the seismic zones along the Nile Valley correspond to its structural configuration (Fig. 1). In southern Egypt, two types of

earthquakes occur in the Aswan Area: natural earthquakes due to the activity of the Kalabsha Fault in the southwest of Aswan, and induced earthquake activity corresponding to the artificial Aswan Lake.

Therefore, the seismic activity in Egypt and its surroundings is controlled by many factors. One of our objectives is to determine the relationship between earthquake activity and the density structure of the crust and upper mantle, which are directly related to active geodynamics in this region.

## 3 Method and initial data

### 3.1 General modelling approach

The integrative analysis of the gravity and other geophysical data follows the procedure that was developed and applied before for Europe, North America, and some parts of Asia (Kaban et al., 2010, 2014, 2015, 2016). This approach implies the following steps:

1. Construction of the initial model of the crust based on available seismic and geological data. This procedure for irregularly distributed data is extensively discussed in Stolk et al. (2013). The crustal model includes at least two layers, sediments and crystalline crust, which are characterized by horizontal and vertical variations of the seismic velocities and density. The densities of the sediments and crystalline crust are determined in a different way. For sediments, we define several types of basins, from "soft" to "hard", which are characterized by different density–depth relations. These relations are determined

based on borehole, compaction, and seismic data (Stolk et al., 2013). The density of the crystalline crust is determined from empirical relationships with seismic velocities (Christensen and Mooney, 1995).

2. The gravity effect of the crust is computed and removed from the observed gravity field. In addition, we remove the effect of deep mantle (below 325 km) heterogeneity based on existing global dynamic models (Kaban et al., 2014b, 2015). The residual gravity anomalies mainly represent the effect of the uppermost mantle and density anomalies of the crust not

included in the initial model, with other uncertainties of the crustal parameters. In the same way, we calculate the residual topography that represents the part of the observed topography/bathymetry, which is not compensated for by crustal density variations including the Moho Boundary. Both these parameters depend on upper mantle density variations but in essentially different ways, which provides the possibility to resolve the vertical density structure (Kaban et al., 2015).





3. To study of the upper crust and evaluate the isostatic state of the lithosphere, high-resolution local isostatic anomalies are computed (Kaban et al., 2016b).

4. The initial 3D density model of the upper mantle is created based on available seismic models. The velocity-to-density conversion factor is computed based on mineral physics relations (e.g., Tesauro et al., 2014).

5. The residual mantle gravity anomalies and residual topography are jointly inverted to estimate the 3D density variations in the upper mantle. The inversion is constrained by the initial model (step 4); the corrections should be minimal. This way, a stable and unique solution can be found. The joint inversion of the residual gravity and residual topography provides the possibility to resolve the vertical density stratification much better than the inversion of the residual gravity anomalies alone because the residual gravity and topography depend on the density heterogeneity but in fundamentally different ways,

depending on the size and depth of the density anomalies. A clear example is a vertical dipole density structure, which is characterized by near zero residual topography however, the gravity field anomaly is very distinctive (Kaban et al., 2015). Compared with previous studies, we also adjust the Moho Boundary in the inversion in places that are not defined well by seismic data.

More details and the computational setup will be described in the following parts.

**3.2 Initial gravity field**

The initial gravity field (free air gravity disturbances) is based on the combined satellite-terrestrial model EIGEN-6c4 (Förste et al., 2014), Fig.2. Maximal resolution corresponds to 2190 spherical harmonic degree/order (~10 km spatially), however the actual one depends on the terrestrial observations included in the model. The long-wavelength part, which is constrained by satellite data (chiefly GRACE and GOCE), is limited to degree/order 240. This resolution is sufficient for modelling of

the upper mantle structure since maximal resolution of the initial data doesn't exceed 1°x1°. However, for computation of the local isostatic anomalies the full gravity field is employed.



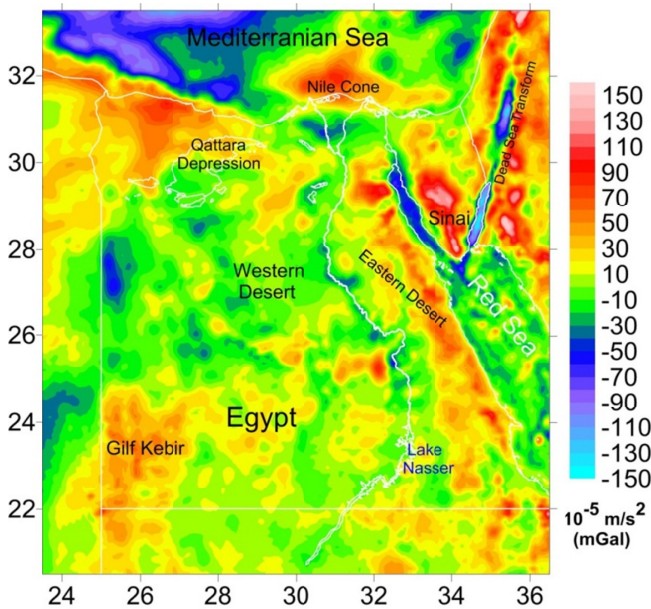

**Figure 2.** Free air gravity disturbances from the combined satellite-terrestrial gravity model EIGEN-6c4 (Förste et al., 2014).

### 3.3 Model of the crust

We use the EPcrust model (Molinari and Morelli, 2011) as a basis for the western part of the study area and the crustal
5   model of Kaban et al. (2016) for the eastern part. These models have been improved by using several regional datasets. The
detailed map of Rybakov and Segev (2004) has been employed for the position of the basement in the northern part of the
study area. The resulting thickness of sediments is shown in Fig. 3a. Density of sediments was estimated according to the
average density-depth relationship (Fig. 3b), which is based on compilation of various data-sets taking into account density-
compaction by Stolk et al. (2013) and then adjusted for the regional data by Kaban et al. (2016). This relationship reflects
10   only a regional trend; therefore small-scale residual anomalies still reflect local density heterogeneity of the sedimentary
layer.





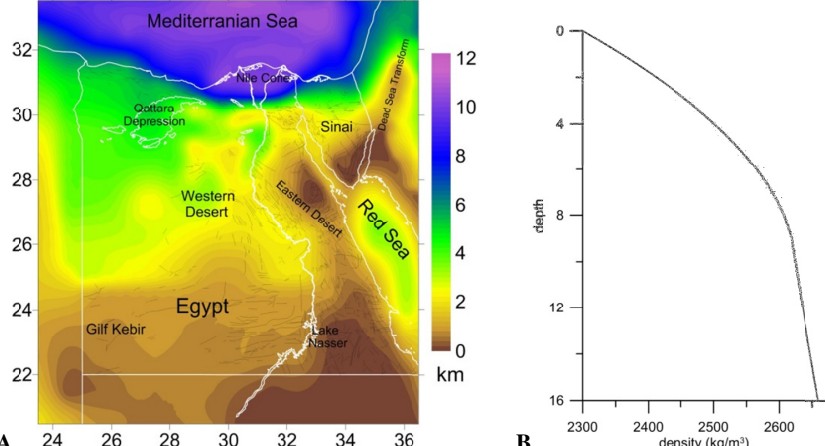

**Figure 3. (A):** Thickness of sediments for Egypt with surroundings. Gray lines show faults. **(B):** Density-depth relationship used for calculation of the gravity effect of sediments.

The Moho model is verified by using original seismic determinations. Most of them are taken from the database of the US
Geological Survey (Mooney, 2010, with updates until April 2017). In addition, the recent receiver function determinations of Hosny and Nyblade (2016) are included (Fig. 4a). For interpolation we used a remove-compute-restore technique developed earlier by Stolk et al. (2013). At the first stage, the measured Moho depths were corrected for the Airy type of isostasy by employing the surface load, which includes the topography/bathymetry and density heterogeneity of sediments. Here, a type of the isostatic compensation is not of primary importance since this correction is restored at the last stage. As demonstrated
by Stolk et al. (2013), the residual Moho values show much less variations than the original ones and can be easily interpolated. After interpolation with the ordinary Kriging technique, the isostatic correction was restored. Therefore, the resulting Moho map fully fits to the original determinations, but demonstrates much better correspondence to the tectonic features than it would be for a direct interpolation. In this way, for example, it is possible to trace extended topography features like the Red Sea in the study are, which are measured only in limited places. This map was then merged at the
boundaries with above-mentioned basic models (Fig. 4a). The final results will be discussed taking into account distribution of the primary data points.

For the crystalline crust, the average P-wave seismic velocities were determined. For this, the s-wave vertical profiles of Hosny and Nyblade (2016) were converted to p-wave velocities using the Vp/Vs ratios provided. The limited amounts of seismic determinations do not provide the possibility to construct a multilayer model; however, for the gravity calculations,





average values are sufficient to estimate the cumulative effect of the crystalline crust (Kaban et al, 2016a). The interpolated data were merged with basic regional models (Fig. 4b).

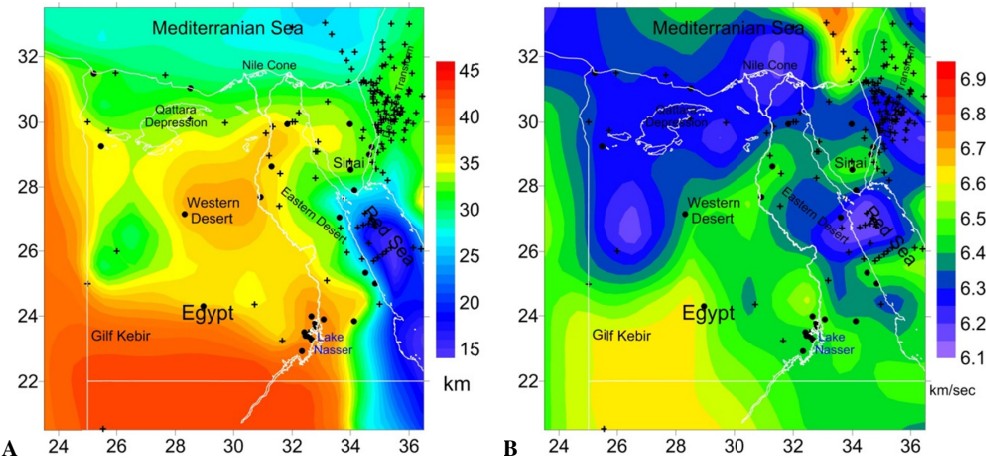

**Figure 4. (A).** Depth to the Moho. Crosses show crustal determinations from the data-base of the US Geological Survey
(Mooney, 2010, with updates up to April, 2017) and circles – the receiver function results of Hosny and Nyblade (2016). **(B)**
Average p-wave velocities in the crystalline crust. Crosses show determinations from the data-base of the US Geological
Survey (Mooney, 2010, with updates up to April, 2016) and circles – the receiver function results of Hosny and Nyblade
(2016). The last ones have been converted from s-wave velocities.

The variations of the p-wave velocities show the fragmentation of the crystalline crust in Egypt and surrounding areas. The
central and northern parts of Egypt and the Sinai massif are characterized by nearly normal velocities (6.4–6.6 km/s), which
are typical for the continental crust (Christensen and Mooney, 1995). The velocities in the northern and northwestern parts
significantly decreased to 6.1–6.3 km/s; the same is true for parts of the Western Desert and the North Red Sea. We also
observe a W–E trend from low to high velocities in the Mediterranean, which likely corresponds to the transition from the
oceanic to continental crust (Fig. 4). The maximum around Gilf Kebir is not well defined because it is based on two marginal
determinations only. The velocities of the crystalline crust were converted into densities by employing the nonlinear
relationships of Christensen and Mooney (1995).

### 3.4 Initial density model of the upper mantle

The initial density model of the upper mantle is based on the tomography model of Schaeffer and Lebedev (2013), which is
converted to densities by applying the mineral physics method of Stixrude and Lithgow-Bertelloni (2005). A complete





description of this technique can be found in (Tesauro et al., 2014; Kaban et al., 2016a). Two slices of density variations for the depths of 75 km and 150 km are shown in Fig. 5. The model demonstrates general trends in the area; the density mainly decreases towards the Red Sea at a depth of 75 km in W–E direction (Fig. 5A). At a greater depth, a strong positive anomaly associated with the subducting African lithosphere appears in the north (Fig. 5B).

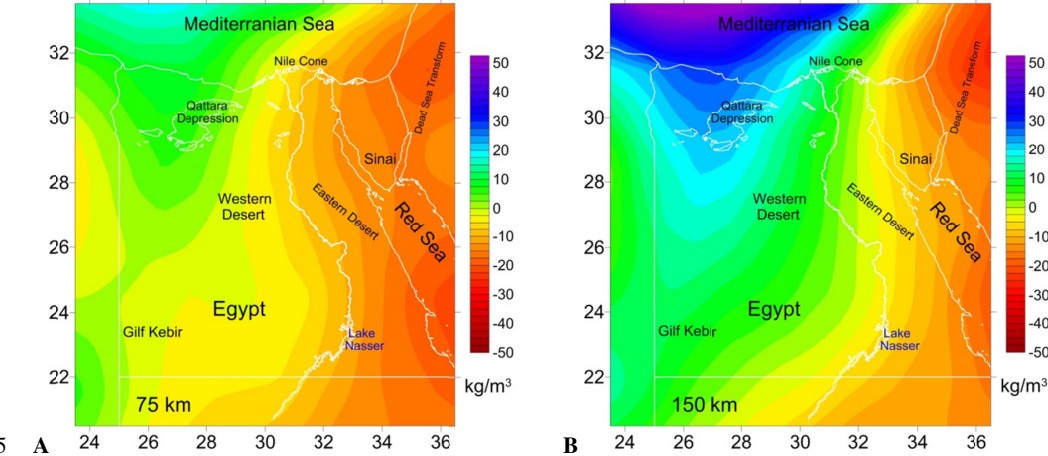

**A**

**Figure 5.** Initial density model of the upper mantle based on the tomography model of Schaeffer and Lebedev (2013). **(A):** depth 75 km; **(B):** depth 150 km.

## 4 Results

### 4.1 Residual mantle gravity anomalies and residual topography

The gravity effect of the crust was determined based on the constructed crustal model. All calculations are performed relative to a reference density model (Table 1). The density of the crustal layers corresponds to the estimates of Christensen and Mooney (1995). The mantle densities are determined as the global averages estimated based on the seismic velocities provided by the model of Schaeffer and Lebedev (2013), assuming a 'fertile' composition of the upper mantle material (Tesauro et al., 2014). The parameters of this model are the same as those used in studies of other regions and on larger

scales (e.g., Kaban et al., 2016), which provides the possibility for direct comparison of the results. It should be noted that the parameters of the reference model are not critical for the results since they mainly affect the average level of the computed field, which is not considered in this study (Mooney and Kaban, 2010).





Table 1. Reference density of the crustal and upper mantle layers.

|  | Upper crust | Lower crust | Uppermost mantle | |
|---|---|---|---|---|
| Depth (km) | 0–15 | 15–40 | 50 | 100 |
| Density (kg/m$^3$) | 2700 | 2940 | 3357 | 3384 |
|  | Upper mantle | | | |
| Depth (km) | 150 | 200 | 250 | 300 |
| Density (kg/m$^3$) | 3419 | 3457 | 3510 | 3560 |

Because the effect of deep layers strongly depends on remote areas and might represent significant trends (even between the northern and southern hemispheres), the high-resolution model of Egypt including the surrounding areas has been embedded in the global model. For this purpose, we used CRUST1.0, which was improved for North America and Eurasia based on recent models for these continents (Mooney and Kaban, 2010; Stolk et al., 2013). The details of the computational technique are described in Kaban et al. (2016b). The residual anomalies obtained by removing the crustal effect from the observed gravity field, are shown in Fig. 6A. The resolution of this field is limited to 1º × 1º, since the resolution of the crustal model don't provide more details even in the places with dense seismic observations (Fig. 4). As already mentioned, we also removed the effect of the deep mantle heterogeneity below 325 km based on a global dynamic model described in Kaban et al. (2015, 2016a).

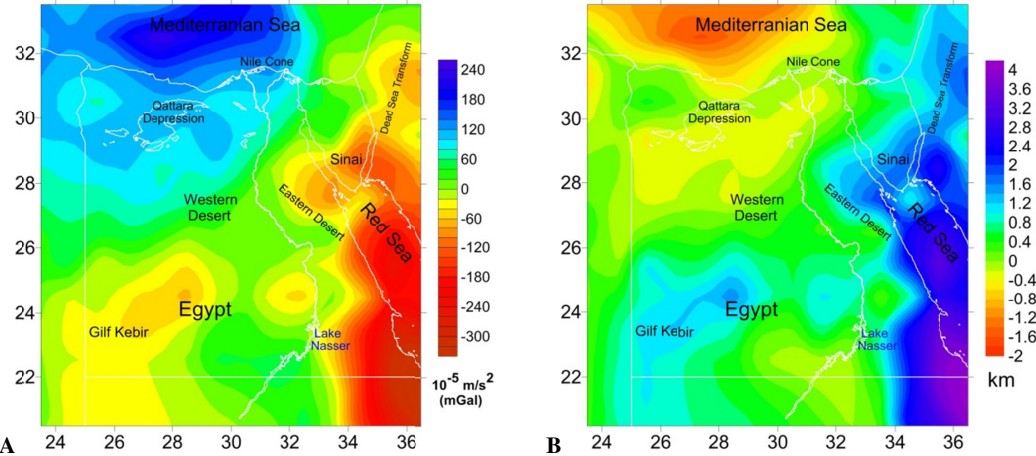

**Figure 6.** Residual mantle gravity anomalies **(A)** and residual topography **(B)** calculated by removing of the crustal and deep mantle fields from the observed gravity.



In addition to the residual gravity anomalies, the residual topography was also computed based on the same crustal model (Fig. 6B). The residual topography acts as load, which is not compensated for by crustal density variations including the Moho.

$$t_{res} = \frac{1}{\bar{\rho}}(\rho_{top})t_{obs} + \frac{1}{\bar{\rho}}\int_{0}^{M}\Delta\rho(h)dh, \qquad (1)$$

where $\rho_{top}$ is the average density above sea level (including sediments and ice); $t_{obs}$ is the topography (zero offshore); $\bar{\rho}=$ 2670 kg/m$^3$, the standard density; $\Delta\rho(h) = \rho - \rho_{ref}$ is the relative density below sea level including water; $h$ is the depth from sea level; and $M$ is the depth to the Moho (below Moho $\Delta\rho(h) = 0$). The dynamic effect of the mantle below 325 km was removed from the residual topography in the same way as for the residual gravity.

Potential uncertainties of the residual field were analyzed in detail in Mooney and Kaban (2010) and Kaban et al. (2016a).
They conclude that for relatively extended anomalies, which are based on several crustal determinations, the uncertainty of the residual gravity should not exceed ~40×10$^{-5}$ m/s$^2$ (mGal), which is much less than the total anomaly (-300 to 250×10$^{-5}$ m/s$^2$). Szwillus and Ebbing (2016) provide even smaller values for uncorrelated uncertainties of the crustal model. The corresponding error of the residual topography is ~0.35 km. However, this conclusion only corresponds to the areas with seismic determinations of the crustal structure (Fig. 4). In the following inversion of the residual fields together with seismic
tomography we consider a possibility for further corrections of the initial density model.

The residual fields significantly differ from the previous study of the whole Middle East (Kaban et al., 2016c). This is mainly due to the new data on the crustal structure, which are included in the present model. One can observe a clear division of the area into several distinctive patterns. The northwestern part is characterized by positive residual gravity anomalies, while negative anomalies dominate in the Red Sea, with some extension to the continental part including Sinai
(Fig. 6A). The residual topography generally mirrors the residual gravity; however, they have several principal differences (Fig. 6B). These fields will be used to adjust the density models of the crust and upper mantle in the following sections.

**4.2 Density model of the upper mantle**

The 3D density model of the mantle has been constructed through joint inversion of the mantle gravity anomalies (Fig. 6A) and residual topography (Fig. 6B) constrained by the initial density model based on seismic tomography (Fig. 5). The
inverse problem implies the minimization of the functional:

$$\min\{\|\, A\rho - g_{res}\,\|^2 + k\,\| B\rho - t_{res}\,\|^2 + \alpha\,\|\rho - \rho_{ini}\,\|^2\}, \qquad (2)$$





where $\boldsymbol{A}$ and $\boldsymbol{B}$ are the integral operators converting the densities $\boldsymbol{\rho}$ into gravity and dynamic topography, $\boldsymbol{g_{res}}$ and $\boldsymbol{t_{res}}$ are the mantle gravity anomalies and residual topography, and $\boldsymbol{k = 2\pi G \rho_t}$ is the scaling coefficient normalizing the topography with respect to gravity ($\boldsymbol{G}$ is the gravitational constant and $\boldsymbol{\rho_t}$ is the density of the topography). The regularization condition requires that the calculated density anomalies are close to the initial model $\boldsymbol{\rho_{ini}}$, where $\boldsymbol{\alpha}$ is the damping factor. The inversion

is performed in the spherical harmonic domain. For the dynamic topography $\boldsymbol{t_{dynamic} = B\rho}$, we use a vertical viscosity–depth profile constrained by mineral physics and geodynamic models (Kaban et al., 2015). The technical details and numerical tests proving the resistance of the solution to plausible changes of the inversion parameters can be found in Kaban et al. (2015a, Supplementary Material).

The model setup is the same as that in Kaban et al. (2015, 2016c). The residual fields and initial density model were
extended to the whole Earth, which is required for the decomposition into spherical harmonics. The same global model as described above was used for these purposes. The initial tomography model of Schaeffer and Lebedev (2013) is global; therefore, it was converted to densities using the same approach as in the study area. Density variations were calculated for seven layers with the central depth at 15, 45, 100, 150, 200, 250, and 300 km, respectively. The density perturbations in the upper layer adjust the potential uncertainties of the crustal densities. The anomalies at the depth 45 km in the continental part
were recalculated in the corrections of the initial Moho model using the crust–mantle density contrast from the reference model (Table 1).

The obtained 3D density model based on joint inversion of the residual gravity and topography constrained by the tomography based initial model is shown in Fig. 7. As was mentioned above, the corrections for the initial Moho map were also estimated, which gives a new map shown in Fig. 7A. As expected, the maximal correction (-3.6 km) is calculated in the
northern part of the study area, which is not covered by seismic data. In other areas, it does not exceed ±2 km; this value corresponds to the uncertainty of the seismic determinations. As mentioned above, the correction was applied to the continental part only, where the Moho depth exceeds 30 km. It remains unmodified for the Red Sea and Mediterranean. The calculated density anomalies range from -35 to 50 kg/m3 and significantly differ from the initial density model (Figs 7B–D). Compared to the large-scale model of the whole Middle East, the present results clearly show the fragmentation of the upper
mantle in Egypt. The central part and the Qattara Depression are characterized by an increased density of the mantle, which extends to the Mediterranean maximum at a depth of 100 km (Fig. 7B). At greater depths, the central Egyptian maximum extends to the southeast (Figs 7C, D). The negative anomaly is localized over the Red Sea and some surroundings at the boundary of the Western Desert and Nubian Shield and limited to a depth of ~150 km, disappearing at greater depths. The local positive anomaly corresponds to the Sinai Massif (Figs 7B, C).



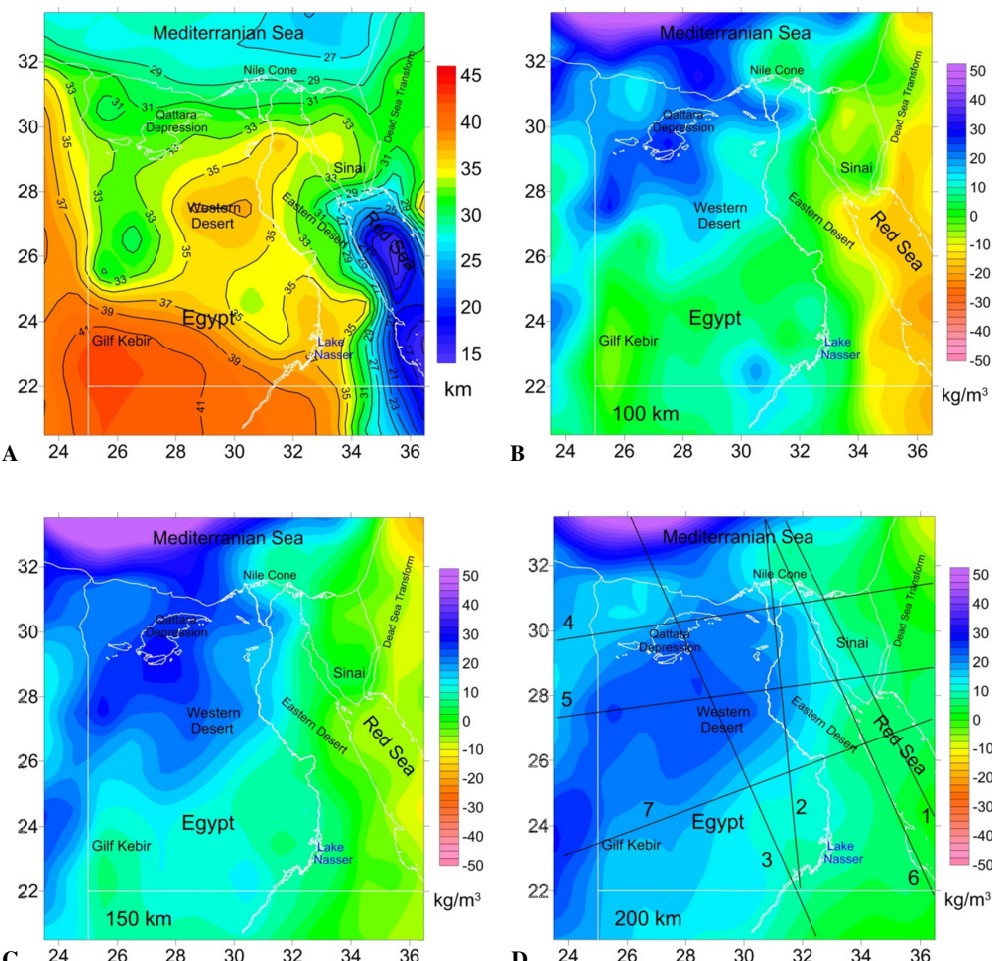

**Figure 7.** Results of the inversion. **(A)**: Corrected Moho Map; **(B-C-D)**: calculated density variations at the depths 100, 150 and 200 km. Location of the profiles in Fig. 8 are shown in the map D.





## 5 Discussion

### 5.1 Fragmentation of the lithosphere in Egypt based on its density structure

We further discuss seven vertical sections along the profiles shown in Fig. 8. Because the vertical resolution is limited to 35–50 km, the anomaly in the crust might be smeared to the uppermost layer of the mantle (> 50 km), especially in offshore

5   areas with thin crust.

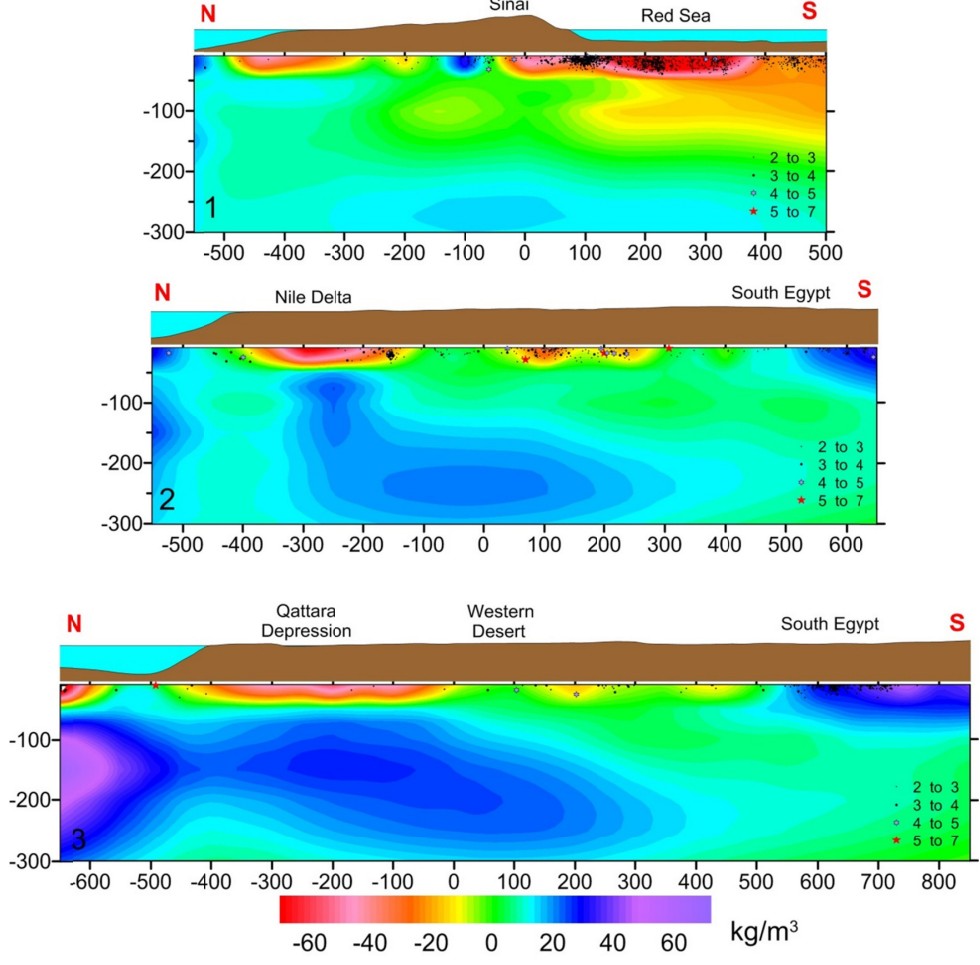

**Figure 8.** Density anomalies along selected profiles. Sections 1-3, Fig. 7D. Black dots indicate the hypocentres of the

10   earthquakes projected on to the cross sections (ENSN earthquake Catalogues).





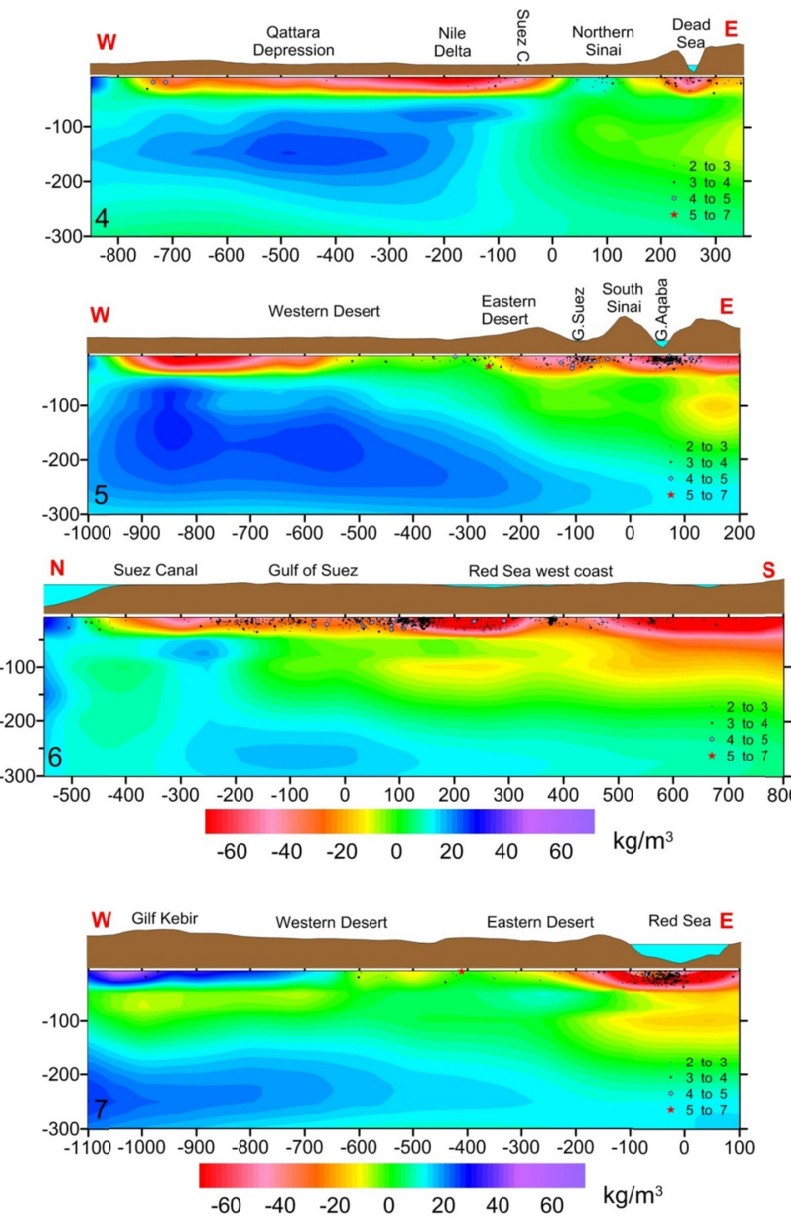

5    **Figure 8 (continue).** Density anomalies along selected profiles. Sections 4-6, Fig. 7D.





The negative density anomaly under the North Red Sea is limited to the uppermost mantle as shown in section 1 (Fig.8), agrees with previous conclusions with respect to the passive origin of the extension in this area (e.g., Bosworth, 2015). This anomaly does not continue to Sinai, which is characterized by neutral or slightly positive densities in the upper mantle. The seismicity is mainly localized in the crust, which is characterized by an extremely low relative density likely related to the

weak layer that is prone to strong deformations, resulting in seismic events. It is also clear that most of the earthquakes are of low magnitude and their hypocenters are concentrated in the crust and confined to the Red Sea Rift. This might be related to the shallow Moho Discontinuity, which is characterized by high temperature and low-density material. Therefore, the stress is not accumulated for a long time. The continuous release of the stress thus generates permanent seismicity characterized by shallow depths and low magnitudes.

The distribution of anomalous density and seismicity from the Nile Delta to South Egypt is clearly indicated in section 2 (Fig.8). It is well known that the Nile Delta is characterized by very low seismic activity compared to the surrounding area (Fig. 1). The constructed density model can provide some explanation for this phenomenon. It is clear from Fig. 8 (section 2) that this area corresponds to a dense and likely strong mantle lithosphere extending to the bottom of the crust. It has been demonstrated that seismicity occurs at the boundaries of rigid lithospheric blocks in similar situations (Tesauro et al., 2015).

In the case of the Nile Delta, the weak crust easily accommodates relatively small deformations, which in contrast to the Red Sea are limited by the strong lithosphere beneath.

The Qattara Depression is also characterized by the high-density lithosphere overlain with relatively low-density crust (Fig. 8, section 3). The high-density zone in the mantle deepens to the North and is localized at depths of 200–250 km in Middle Egypt. In the east, the high-density lithosphere extends to the Suez Line (Fig. 8, section 4), which clearly marks the

boundary between the strong lithosphere in Western Egypt and the weaker lithosphere in the east. This result agrees with estimations of the effective elastic thickness of the lithosphere based on the cross-spectral analysis of the gravity field (Chen et al, 2015). The seismicity behaviour at the edge of the high-density lithosphere block is similar to that at the southern border of the Nile Delta (Fig. 8, section 2). Further to the south, the transition between different lithosphere blocks is smoother; the high-density lithosphere gradually deepens from the Western Desert to Sinai (Fig. 8, section 5).

The crustal and mantle structure along the Suez Canal and Gulf of Suez is shown in section 6 (Fig.8). One of the vital problems discussed for these structures is the cessation of the opening of the Gulf of Suez Rift. It is observed that the boundary between the Gulf and Isthmus of Suez corresponds to the boundary between the lithospheric blocks with different densities (Fig. 8, section 6). The high-density block in the south-eastern Mediterranean located to the north of the Suez Canal and Gulf of Suez might terminate the prolongation of the Gulf of Suez Rift further to the north. It also manifests the

significant decrease of seismicity in the Suez Isthmus and further to the north. Therefore, the stronger lithosphere in the north might prevent the continuation of the Gulf of Suez opening.



The high-density block is observed in the area of Gilf El Kebir in southwestern Egypt, as indicated in section 7 of Fig. 8, with no associated seismicity revealing the stability of this region which is characterized by Paleozoic outcrop. A low-density anomaly can be observed in the North Red Sea associated with high earthquake activity (Section 7, Fig.8). The lithosphere structure becomes asymmetric across the Northern Red Sea Rift, which corresponds to the asymmetric pattern of

the seismicity relative to the central axis of the North Red Sea. The intensive seismicity is concentrated on the western side, where the lithosphere is weakened at a depth of 100 km (section 7, Fig. 8). Further to the north, the seismicity pattern is divided into two branches (Fig.2). The Red Sea Zone continues in the direction towards the Gulf of Suez (Fig. 1). Another branch extends along the Dead Sea Transform Fault, which is also characterized by high seismic activity. To conclude, the seismicity asymmetrically tends to the west of the North Red Sea Rift, possibly because the opening of the North Red Sea

Rift is directed W–N–W to the Gulf of Suez, which is associated with the weakened lithosphere.

**5.2 Isostatic gravity anomalies and their relation to seismicity.**

The local isostatic anomalies image upper crust density heterogeneities, which are not included in the initial model, in particularly not completely compensated in local isostasy sense but rather supported by the rigid lithosphere. They can be generated by various processes (e.g., intrusion of mantle batholiths in the upper crust, faulting, and subduction) and could be

associated with significant stresses in the lithosphere. Therefore, isostatic anomalies of the gravity field are often used to study active seismic areas (e.g., Assumpção and Sacek, 2013; Sobiesiak et al., 2007). However, the feasibility of this approach strongly depends on the isostatic model which was used to calculate the isostatic anomalies. The standard simple models (Airy and Pratt) that are based on the observed topography often differ from the real density structure of the crust and upper mantle, which can cause artificial anomalies (e.g., Kaban 2016b). Therefore, it is important to take into account as

much as possible actual information about the crustal structure in the study area, which can be obtained from other geophysical and geological methods.

The isostatic anomalies, which are analyzed in this study, are compiled in the following way. First, we use the residual part of the mantle gravity anomalies (Fig. 6a), which is not fit in the inversion. This is a long-wavelength field, which is characterized by small amplitudes (-9 - +11×10$^{-5}$ m/s$^2$, mGal). This field has been complemented by the local part of the

isostatic anomalies, which were computed in the previous study for a high-resolution grid (Kaban et al., 2016b). The total isostatic anomalies are shown in Fig. 9. Their resolution corresponds to the resolution of the initial gravity field model EIGEN-6c4 (max 10x10 km).





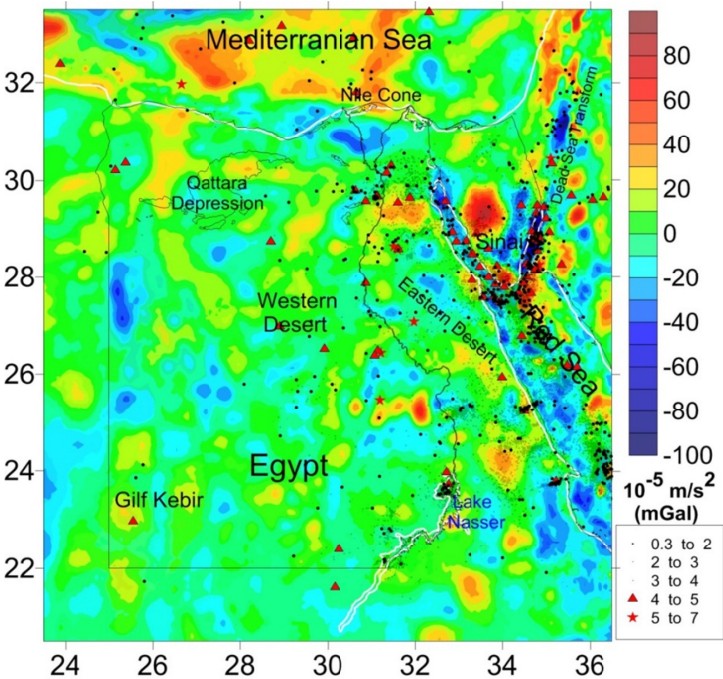

**Figure 9.** Isostatic anomalies of the gravity field and seismicity. Earthquakes with M < 3 are half-transparent to prevent masking of the isostatic anomalies.

The isostatic anomalies demonstrate very diverse patterns in Egypt and its surroundings. The strongest variations ($\pm 90 \times 10^{-5}$

5    m/s$^2$, mGal) are found in the south-eastern part along the Red Sea and Sinai Peninsula (Fig. 9). Sinai is bounded by linear anomalies parallel to the Gulf of Suez and Aqaba Gulf, which are clearly associated with high levels of seismicity.

The whole Sinai Peninsula is divided into several parts with different patterns of the isostatic anomalies. The central block with very high anomalies up to $100 \times 10^{-5}$ m/s$^2$ (mGal) demonstrates very low seismic activity. It is divided from the southern part by a narrow high-amplitude negative anomaly (Fig. 9). The earthquakes tend to occur in zones with high gradients of the

10   isostatic anomalies. This tendency persists to the west over the Nile Valley and Eastern Desert. The seismicity in the Red Sea is also concentrated in the high-gradient zones. This certainly concerns only a part of the earthquakes; there are many other factors controlling seismicity, but the general tendency is clear.

The broad negative anomalies in the western part of the area and over the Qattara Depression likely indicate the increased thickness of the low-density upper crust with no seismic activity associated. In the same way, the broad negative anomaly



over the Nile Delta indicates that the density of sediments is slightly overestimated in the initial model. The high amplitude isostatic anomalies in the Mediterranean might be related to the subduction of the African lithosphere under Eurasia.

**6 Conclusions**

A joint analysis of the new satellite-terrestrial model of the gravity field, and the recent data on the crustal structure and tomography model was performed to create an integrative model of the crust and upper mantle and to investigate the relationship between the isostatic state of the lithosphere and seismicity. The following conclusions can be drawn.

1. This study reveals a distinct fragmentation of the lithosphere of Egypt into several blocks which are characterized by different properties.

2. The central area and the Qattara Depression are characterized by an increased density of the mantle, which extends to the Mediterranean maximum at a depth of 100 km. At the same time, the crystalline crust in this area demonstrates low average seismic velocities and density, which might indicate an increased thickness of the relatively low-density upper crust.

3. The central Egyptian maximum of the upper mantle density extends to the southeast in the mid–upper mantle and is localized at depths of ~170–270 km. The same trend is found in W–E direction; however, it is limited by the western part of the Eastern Desert.

4. In the northeastern part of Egypt, the high-density lithosphere is bounded by the Gulf of Suez, which marks the transition between the typically strong and cold plate and the weakened lithosphere.

5. The Sinai Microplate is characterized, on average, by the normal density of the upper mantle; however, smaller-scale features cannot be resolved at these depths because they are already smoothed out in the gravity field. In contrast, the upper crust, which is imaged by the isostatic anomalies, demonstrates large density variations. The central block has a strong maximum, which should correspond to the strong and dense crust.

6. The density structure of the lithosphere is closely related to the seismicity distribution. The low seismicity in the Nile Delta and Suez Canal might be related to the increased strength of the lithosphere, which is associated with densification due to low temperatures. It prevents strong deformations; the weak crust accommodates insignificant strains. In the same way, the increased strength of the lithosphere in the Suez Isthmus and further to the north prevents the Suez Gulf from opening further.

7. The negative mantle anomaly in the North Red Sea is limited to the uppermost mantle, which confirms the passive origin of this structure. The low-density and likely weak upper crust and uppermost layer of the mantle are characterized by high seismic activity.

8. The density structure of the lithosphere in the northern Red Sea is asymmetric; the western side is characterized by low densities at a depth of ~100 km, which likely corresponds to the hot weakened layer. Most earthquakes are of low magnitudes at shallow depths and are located to the west of the axial depression of the Red Sea Rift. Thus, earthquakes are confined to the crust and uppermost mantle, where the low strength provokes stress release.

9. The continuation of the Suez Rift further to the north might be blocked by the strong lithosphere in the northern direction.

10. We found a correlation between the variations of the isostatic anomalies and seismicity. High-amplitude and localized isostatic anomalies generally correspond to areas with high seismic activity. This tendency is especially visible in Sinai, which is bounded by strong linear isostatic anomalies with a corresponding increase of seismic activity. Less pronounced but still visible, this relationship extends to the west including the Nile Valley and Western Desert and to the North Red Sea.

**Data availability**

The gravity data are available from the International Centre for Global Earth Models (ICGEM) (http://icgem.gfz-potsdam.de/home). The tomography model of Schaeffer and Lebedev (2013) is available from the website of the author (https://andrewjschaeffer.wordpress.com/tomography/sl2013sv/). The data on the crustal structure are available by contacting the corresponding authors.

**Author contribution**

MKK and SEK conceived the study. MKK carried out the gravity data analysis and construction of the 3D density model. SEK and NAA carried out the analysis of seismicity and its relation to the lithosphere density distribution and isostatic anomalies. MKK, SEK and NAA contributed to interpretation of the results and conclusions and wrote the manuscript.

**Competing financial interests**

The authors declare no competing financial interests.





**Acknowledgements**

The authors are grateful to the International Scientific Partnership Program ISPP for funding this research work through ISPP#0052.

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
