# Peer review of "Density structure and isostasy of the lithosphere in Egypt and their relation to seismicity"

_Solid Earth, 2018_

## Referee Comment (RC1) · A.N. Minakov (Referee) · 9 Apr 2018

Mikhail Kaban and colleagues present in their paper an interesting study linking the seismicity distribution, mantle density structure and isostasy in Egypt and the southeastern Mediterranean region. They compile an extensive database of controlled-source and passive seismology data to constrain the crustal model. The conversion of global shear wave velocity model for the mantle is done using mineral physics constraints. The starting density model is further improved using the inversion of both gravity anomalies and residual topography. Their results show that the dense lithosphere in northern Egypt corresponds to a low-seismicity region whereas the less dense lithosphere in the northern Red Sea and the Gulf of Aqaba are more seismically active. The authors also find an intersting relation between isostatic anomalies and distribution of

seismicity.

The presentation of the paper can be improved. The first-order structure of the lithosphere: the regions of continental cratonic and extended/oceanic lithosphere is not easy grasp from the figures. The location of plate boundaries and continent-ocean boundaries would be very useful to show in the figures (both in Red Sea and Mediterranean). Would useful to emphasize which lithospheric plates are involved (Africa, Arabia, Sinai. . .). The figures can be improved. The small symbols for earthquakes are hardly seen (both in maps and cross-sections). Perhaps, zoomed plots for the seismically active regions can be included. The density perturbation plots are a bit confusing. Perhaps, a couple of transects with absolute densities and seismic velocities can be shown. Could the location of transects located be added to the maps showing the distribution of seismic events?

Detailed comments to address for improving the paper:

Page 2. Line 2. Âń..compositional variationsÂż in the mantle. What about compositional variations in the crust vs temperature Line 12. Which studies: controlled-source, ambient noise etc. please, detail. Line 25. "..satellite and terrestrial data" including land areas (complementary to satellite radar altimetry). Line 30. "1-2 parameters" what are these paramteres? Thicknesses, densities? Line 31-32. "..gravity approach". Do you mean inversion? Page 3. Line 1. "entirety" do you mean entire? Line 5. "marginally touches" ? do you mean "partly covers"? Line 6. "low seismicity in northern Egypt..". Why does it appear anomalous? Please, explain. Line 9. "shear zones". Where are these shear zones located? Hardly can be seen in the figures..Please, show these shear zone more in the figure. Figure 1 can be improved to make visible earthquakes and faults. Line 24 Do you mean Arabian Plate? Please, detail.. Page 4. Figure is very busy. Perhaps, presenting zoomed northern Egypt would be useful. Please, show more clearly shear zones.. What are the "principal trends" of plate motion? Page 5 Line 23. "existing global dynamic models". Which one is used in this study? Page 6. How do you find the isostatic topography? Do you do iterations? Do you have analytic

formula? Page 7. Please, add COB and location and type of plate boundaries in the figure. Page 8. "p-wave" velocity, "P-wave" velocity or "Vp" ? please, choose one. Page 9. Location of seismic determinations are confusing because of association with seismicity distribution. Could you improve it? Is the interpolation/extrapolation of crustal thickness based on singular measurements (e.g. southern part of figure)? Perhaps, would be necessary to blank the area beyond certain search radius of interpolation. Could you add an uncertainty estimate from kriging? Line 18. "initial density model". Sometimes absolute densities and density perturbations are interchanged in the text and formulas. Could you make it clear what you are talking about in each particular case? Page 10. Line 17. The absolute densities would be important for computation of residual topography. Please, detail. Page 11. Line 7. 1x1 degree resolution. What do you mean? Grid cell size? Line 9. Why 325 km depth? Please, explain why you chose with depth as a lower limit of the model. Page 12. Line 6. "rho_ref". Does it refer to Table 1? Please, comment on the application of this formulation to oceanic domains.. Page 13. What is the difference between "t_res" and "t_dyn" do you use the different "B" operators to compute them. Do you obtain isostatic topography using the compensation depth of 325km given mantle density model? Do you iterate? Please, detail..

Line 14. "The anomalies at the 45 km depth". What kind of anomalies (not clear)? Gravity anomalies? Page 14. Can you show a difference plot between starting and final density model? This would be very useful to appreciate the inversion results. How much the initial model was updated comparing various regions? Page 15. Please, show a profile with absolute density/seismic velocity to better present the lithospheric structure. Please, show the location of transects and epicenters on the same map. Line 3. "vertical resolution". What resolution you are talking about? Do you have a reference for that? Page 17. Line 3. "neutral or slightly positive densities". Better small positive density anomalies. "Shallow Moho discontinuity..material". Do you mean this material is mantle rocks and located below the Moho? Line 12. Please, replace "section" to "Profile" to denote transects in the text. Otherwise, to me it is confusing

with the manuscripts sections. Line 31. Reference to "Steckler, M. and U. ten Brink 1986. Lithospheric strength variations as a control on new plate boundaries: examples from the northern Red Sea. Earth and Planetary Science Letters, v. 79, nos. 1 and 2, p. 120-132" would be useful here.

Page 18. Line 1-3. What does it low and high density anomalies reflect? Temperature, different composition? Please, explain. Line 14. "mantle batholiths in the upper crust". Do you mean granite batholiths or mantle plumes? Please, explain.

Line 17-18. "Standard simple models . . . differ from the real density structure". "Models" and "structures" not exactly comparable things.. Line 23. "long-wavelength FIELD" do you mean gravity anomalies? What are these wavelengths that you are considering long? Line 24. Do you mean about 10 mGal variation? Page 19. Figure 9. Symbols are too small to be seen. Leave just "mGal" for colorbar. The plot is very busy the symbols are masked by the color of the background.

---

## Referee Comment (RC2) · B. Root (Referee) · 17 Apr 2018

The authors present a regional study of the lithosphere underneath Egypte. The density structure is studied by combining gravity data with other geophysical information of the crust and upper mantle. The motivation of the study is to see if there is a relation between the observed seismicity and the density structure of the subsurface. I find this an interesting approach and application for the presented gravity field modelling done in the study. And in my opinion this relation could even be relatively more addressed. The study is performed with a well documented methodology. I believe this paper could become interesting after some improvements:

My suggestions for improvement:

[Figure]

R1: Transparency of data and methods. I find the data and models used should be more explicitly discussed. This would give a better understanding of the robustness of the presented model. on page 5 line 19-20, data from Stolk et al. (2013) is used. But what data is this in Egypte (location, value, uncertainty.) page 6 line 4 "available seismic models" -> which models? page 7 line 5 "...several regional datasets" -> which ones and how did this affect the model? line 8 "... various data-sets..." similar questions? page 11 line 9-10: which seismic model is used for the deep Earth and how does it relate to Schaeffer and Lebedev (2013) model? Is it compatible? And by removing the deep Earth, is only the gravity field of the mantle anomalies removed or also the dynamic signal due to the mantle convection? How is this related to your later isostasy study? page 13 line 6-8 "The technical details... (2015a, Supplementary)", could this be described in a few line, such that the reader does not have to go to this other literature. It will improve the readability of the manuscript. Page 19 line 11-12 "many other factors controlling seismicity" -> which are?

R2: Overall, I find the authors could elaborate and discuss more on their findings, because this is the most interesting part of the paper. Some examples: page 12 line 20 "however, they have several principal differences" -> which are, please discuss them. page14 Figure 7 shows the densities in the upper mantle. I miss the discussion between figure 5, where also densities of the upper mantle are shown. Why are there differences and what can they teach use about the subsurface. And maybe to keep the comparison fair, similar depths and wavelength bandwidths should be used, because now it is difficult to compare the quantitative differences. Could the differences tell use about different compositions in the upper mantle?

R3: One of the most interesting issues is the relation of the seismicity to the (non-)isostasy. I find this should get more attention in the manuscript, only after page 18 I read about it. One of the conclusions is (page 19 lines 7-13) is that seismicity occurs in zones with high gradients. Would it be better to plot the gravity gradients and find out? Maybe use invariant of gradients, this would remove the reference frame

dependencies, or another method? It would back-up this conclusion.

Minor comments:

m1 table 1: might be better to use a graph, because than it is better to compare to other literature that uses graphs. Or might be good to use both.

m2: Figure 2, please add to what degree and order is used in this figure in the caption.

m3 Figures with longitude and latitude: I miss the labels in many of the maps.

m4 Figure 8: Why was the Moho not inserted in this figure. It would be a good addition to the cross-sections and give the reader a better understanding of the constructed model.

m5: page 5 line 21: how does the uncertainty in the empirical relationships of Christensen and Mooney 1995 affect your results?

m6: textual detail: page 13 line 18 and 21 "as mentioned above" was used twice. Also, this is a bit ambiguous, is it about the sentence, section, or whole paper above this line.

m7: page 18 line 23: "long-wavelength fileld" -> field. Nad what is meant with long-wavelength, specify with d/o.
* * *

---

## Editor Comment (EC1) · M. Mandea (Editor) · 25 Apr 2018

Based my own reading of the manuscript and my experience in the field, I found that this contribution might be of interest to the journal. However, important revisions are required before its publication. The main issue is the robustness of the interpretation, considering that little is indicated about the accuracy of used data and robustness of the models (which, by the way, need to be explicitly described). On the other hand, some statements are too vague (e.g. "various data-sets", "many other factors", etc), and they have to be clearly denoted. Some more information about the regional setting can be obtained from world digital magnetic anomaly map: this kind of data could be important to correlate with the gravity anomalies.

[Figure]

Efforts are needed to improve the figures, many being difficult to read.

---

## Author Comment (AC1) · 9 May 2018

**Response to the comments of the 1ˢᵗ reviewer (Dr. Alexander Minakov).**

The original comments are in italic.

*Mikhail Kaban and colleagues present in their paper an interesting study linking the seismicity distribution, mantle density structure and isostasy in Egypt and the southeastern Mediterranean region. They compile an extensive database of controlledsource and passive seismology data to constrain the crustal model. The conversion of global shear wave velocity model for the mantle is done using mineral physics constraints. The starting density model is further improved using the inversion of both gravity anomalies and residual topography. Their results show that the dense lithosphere in northern Egypt corresponds to a low-seismicity region whereas the less dense lithosphere in the northern Red Sea and the Gulf of Aqaba are more seismically active. The authors also find an intersting relation between isostatic anomalies and distribution of seismicity.*

We are grateful to the reviewer for the positive evaluation of our work. His comments are very useful and help us to improve the manuscript.

*The presentation of the paper can be improved. The first-order structure of the lithosphere: the regions of continental cratonic and extended/oceanic lithosphere is not easy grasp from the figures. The location of plate boundaries and continent-ocean boundaries would be very useful to show in the figures (both in Red Sea and Mediterranean). Would useful to emphasize which lithospheric plates are involved (Africa, Arabia, Sinai. . .). The figures can be improved. The small symbols for earthquakes are hardly seen (both in maps and cross-sections). Perhaps, zoomed plots for the seismically active regions can be included. The density perturbation plots are a bit confusing. Perhaps, a couple of transects with absolute densities and seismic velocities can be shown. Could the location of transects located be added to the maps showing the distribution of seismic events?*

The figures will be improved according to the reviewer's suggestions. In the revised manuscript we will demonstrate locations of the main plate boundaries in the study area. We will also improve visibility of the earthquakes in all figures. The absolute densities will be shown instead of density perturbations in Figs. 5 and 7.

*Detailed comments to address for improving the paper:*

*Page 2. Line 2. ´n..compositional variationsÂ˙z in the mantle. What about compositional variations in the crust vs temperature Line 12. Which studies: controlled-source, ambient noise etc. please, detail.*

This is described in details in the section 3.3 "Model of the crust".

*Line 25. "..satellite and terrestrial data" including land areas (complementary to satellite radar altimetry).*

This clarification is added.

*Line 30. "1-2 parameters" what are these paramteres? Thicknesses, densities? Line 31-32. "..gravity approach". Do you mean inversion?*

Yes, this is clarified in the revised manuscript.

*Page 3. Line 1. "entirety" do you mean entire? Line 5. "marginally touches" ? do you mean "partly covers"?*

Correct, this is clarified.

*Line 6. "low seismicity in northern Egypt..". Why does it appear anomalous? Please, explain.*

One of the main goals of this study is to explain the anomalously low seismicity in northern Egypt.

*Line 9. "shear zones". Where are these shear zones located? Hardly can be seen in the figures..Please, show these shear zone more in the figure. Figure 1 can be improved to make visible earthquakes and faults.*

This figure is improved according to the reviewer's suggestions.

*Line 24 Do you mean Arabian Plate? Please, detail..*

Yes, this is clarified.

*Page 4. Figure is very busy. Perhaps, presenting zoomed northern Egypt would be useful. Please, show more clearly shear zones.. What are the "principal trends" of plate motion?*

Following the reviewer's suggestion, we demonstrate a zoomed figure for northern Egypt.

*Page 5 Line 23. "existing global dynamic models". Which one is used in this study?*

This is clarified in the following parts of the paper.

*Page 6. How do you find the isostatic topography? Do you do iterations? Do you have analytic formula?*

Here we mention the isostatic gravity anomalies. Their computation is described in the following parts.

*Page 7. Please, add COB and location and type of plate boundaries in the figure.*

The figure is improved based on the reviewer's suggestions.

*Page 8. "p-wave" velocity, "P-wave" velocity or "Vp" ? please, choose one.*

P-wave velocity is used as a definition for the term Vp. This is clarified.

*Page 9. Location of seismic determinations are confusing because of association with seismicity distribution. Could you improve it? Is the interpolation/extrapolation of crustal thickness based on singular measurements (e.g. southern part of figure)? Perhaps, would be necessary to blank the area beyond certain search radius of interpolation. Could you add an uncertainty estimate from kriging?*

We clarify that these are seismic determinations of the crustal structure. It is difficult to estimate the overall accuracy of the Moho map because it is based on several existing models and even the accuracy of the existing seismic determinations is undefined. It is used as an initial approximation, which is adjusted then in the inversion.

*Line 18. "initial density model". Sometimes absolute densies and density perturbations are interchanged in the text and formulas. Could you make it clear what you are talking about in each particular case?*

We have clarified this issue.

*Page 10. Line 17. The absolute densities would be important for computation of residual topography. Please, detail.*

We agree that the residual topography depends on the absolute densities. However, the reference model chiefly influences the average level of this parameter, which is not interpreted in this study.

We consider only variations of the residual topography, which are less sensitive to absolute densities. This is clarified in the revised manuscript.

*Page 11. Line 7. 1x1 degree resolution. What do you mean? Grid cell size? Line 9. Why 325 km depth? Please, explain why you chose with depth as a lower limit of the model.*

Yes, this is the grid cell size. The 325 km depth is chosen based on our previous studies as a depth, which exceeds the maximal depth of the lithospheric roots. This is important since the inversion is performed globally. This is clarified in the revised manuscript.

*Page 12. Line 6. "rho_ref". Does it refer to Table 1? Please, comment on the application of this formulation to oceanic domains..*

In the description of the Eqs. 1 it is specified that " rho – rho_ref" is the relative density below sea level including water, which means that rho=rho_water at corresponding depths in the ocean.

*Page 13. What is the difference between "t_res" and "t_dyn" do you use the different "B" operators to compute them. Do you obtain isostatic topography using the compensation depth of 325km given mantle density model? Do you iterate? Please, detail..*

The dynamic topography is a part (chiefly long-wavelength) of the residual topography. In the mantle, we are already considering all dynamic effects, which depend on the viscosity of the mantle, but not a simple isostatic column as In Eqs. 1 for the crust. This is clarified in the text. We also add an additional reference to the original papers, where this method was initially introduced and fully tested.

*Line 14. "The anomalies at the 45 km depth". What kind of anomalies (not clear)?*

These are the density anomalies obtained in the inversion. This is clarified.

*Page 14. Can you show a difference plot between starting and final density model? This would be very useful to appreciate the inversion results. How much the initial model was updated comparing various regions?*
*Page 15. Please, show a profile with absolute density/seismic velocity to better present the lithospheric structure. Please, show the location of transects and epicenters on the same map.*

We have demonstrated in Fig. 7 in the revised manuscript the corrections together with the final density variations. In this figure we show absolute densities of the mantle. In the profiles we still keep their perturbations, which are essential for the interpretation. Also, the reference density in each layer is not adjusted in the inversion and therefore is somewhat arbitrary. This is clarified.

*Page 15. Line 3. "vertical resolution". What resolution you are talking about? Do you have a reference for that?*

The vertical resolution is limited by the model set-up in the inversion. This is clarified.

*Page 17. Line 3. "neutral or slightly positive densities". Better small positive density anomalies. "Shallow Moho discontinuity..material". Do you mean this material is mantle rocks and located below the Moho?*

Yes, we mean the material below the Moho. Both statements are changed accordingly.

*Line 12. Please, replace "section" to "Profile" to denote transects in the text. Otherwise, to me it is confusing with the manuscripts sections.*

We have replaced "section" to "profile" in the whole manuscript.

*Line 31. Reference to "Steckler, M. and U. ten Brink 1986. Lithospheric strength variations as a control on new plate boundaries: examples from the northern Red Sea. Earth and Planetary Science Letters, v. 79, nos. 1 and 2, p. 120-132" would be useful here.*

Thank you for this recommendation. We have added this reference.

*Page 18. Line 1-3. What does it low and high density anomalies reflect? Temperature, different composition? Please, explain. Line 14. "mantle batholiths in the upper crust". Do you mean granite batholiths or mantle plumes? Please, explain.*

In the first case, some additional data are required to explain this maximum. To clarify, we have changed "mantle batholiths" to "mantle intrusions".

*Line 17-18. "Standard simple models . . . differ from the real density structure". "Models" and "structures" not exactly comparable things.. Line 23. "long-wavelength FIELD" do you mean gravity anomalies? What are these wavelengths that you are considering long?*

We have changed to "Standard simple models . . .don't adequately describe the real density structure". The boundary wavelength corresponds to the maximum resolution of the density model (1x1 degree), therefore it is equal to approx. 222 km, which is clarified.

*Line 24. Do you mean about 10 mGal variation?*

Correct.

*Page 19. Figure 9. Symbols are too small to be seen. Leave just "mGal" for colorbar. The plot is very busy the symbols are masked by the color of the background.*

The figure is enlarged. We are obliged to keep the SI unit, however also add mGal for clarity.

---

## Author Comment (AC2) · 9 May 2018

**Response to the comments of the 2[nd] reviewer (Dr. Bart Root).**

The original comments are in italic.

*The authors present a regional study of the lithosphere underneath Egypte. The density structure is studied by combining gravity data with other geophysical information of the crust and upper mantle. The motivation of the study is to see if there is a relation between the observed seismicity and the density structure of the subsurface. I find this an interesting approach and application for the presented gravity field modelling done in the study. And in my opinion this relation could even be relatively more addressed. The study is performed with a well documented methodology. I believe this paper could become interesting after some improvements*:

Thank you for the positive evaluation of our study. We appreciate all the comments and will implement them in the revised manuscript.

*My suggestions for improvement*:

*R1: Transparency of data and methods. I find the data and models used should be more explicitly discussed. This would give a better understanding of the robustness of the presented model. on page 5 line 19-20, data from Stolk et al. (2013) is used. But what data is this in Egypte (location, value, uncertainty.) page 6 line 4 "available seismic models" -> which models? page 7 line 5 "...several regional datasets" -> which ones and how did this affect the model? line 8 "... various data-sets..." similar questions*?

Actually, in this section we describe only the „general modelling approach", which is applicable to any region, without specifying particular data sets. All the data are described in the following section 3.3 "Model of the crust". This is clarified. Also, we strengthen discussion of the robustness of the used data-sets in the revised manuscript.

*page 11 line 9-10: which seismic model is used for the deep Earth and how does it relate to Schaeffer and Lebedev (2013) model? Is it compatible? And by removing the deep Earth, is only the gravity field of the mantle anomalies removed or also the dynamic signal due to the mantle convection*?

The Schaeffer and Lebedev (2013) model provides velocities only for the upper mantle. Therefore, for the deeper layers we use the model s40rts of Ritsema et al. (2011). For the upper mantle it well corresponds to the first model, but its resolution is lower. We take into account the dynamic effects of the mantle convection. These issues are clarified.

*How is this related to your later isostasy study? page 13 line 6-8 "The technical details... (2015a, Supplementary)", could this be described in a few line, such that the reader does not have to go to this other literature. It will improve the readability of the manuscript*.

For the isostasy study, we intend to separate the gravity anomalies, which are chiefly related to the density inhomogeneities in the crust, which are not compensated in both ways: via density heterogeneity of the lithosphere or dynamically from the mantle. Therefore, the residual anomalies not adjusted in the inversion represent a large scale part of the isostatic anomalies, which might be responsible for the stress concentration and seismicity. This is clarified.

*Page 19 line 11-12 "many other factors controlling seismicity" -> which are?*

Primarily, these are deformations related to plate motions. This is clarified.

*R2: Overall, I find the authors could elaborate and discuss more on their findings, because this is the most interesting part of the paper. Some examples: page 12 line 20 "however, they have several principal differences" -> which are, please discuss them.*

These differences are chiefly related to different relative amplitudes of the anomalies depending on the depth to the anomalous body responsible for them. This is clarified.

*page14 Figure 7 shows the densities in the upper mantle. I miss the discussion between figure 5, where also densities of the upper mantle are shown. Why are there differences and what can they teach use about the subsurface. And maybe to keep the comparison fair, similar depths and wavelength bandwidths should be used, because now it is difficult to compare the quantitative differences. Could the differences tell use about different compositions in the upper mantle?*

The densities in Fig. 5 represent the initial model of the mantle, and in Fig. 7 – the final model after the inversion. In the revised manuscript we show in Fig. 7 also the corrections to the initial model, which fully demonstrate the changes after the adjustment. In the revised manuscript we show density variations in Fig. 5 at the same depths as in Fig. 7.

*R3: One of the most interesting issues is the relation of the seismicity to the (non-)isostasy. I find this should get more attention in the manuscript, only after page 18 I read about it. One of the conclusions is (page 19 lines 7-13) is that seismicity occurs in zones with high gradients. Would it be better to plot the gravity gradients and find out? Maybe use invariant of gradients, this would remove the reference frame dependencies, or another method? It would back-up this conclusion.*

Thank you for this advice. In the revised manuscript we show a zoomed figure, which demonstrates variations of the maximal value of the horizontal gradient of the isostatic anomalies for Sinai and surrounding area together with the seismicity.

Minor comments:

*m1 table 1: might be better to use a graph, because than it is better to compare to other literature that uses graphs. Or might be good to use both.*

We still prefer to use the table format since it is important to see exact values of the reference densities. An additional graph would be excessive to our opinion.

*m2: Figure 2, please add to what degree and order is used in this figure in the caption.*

Added.

*m3 Figures with longitude and latitude: I miss the labels in many of the maps.*

They have been added to the figure captions.

*m4 Figure 8: Why was the Moho not inserted in this figure. It would be a good addition to the cross-sections and give the reader a better understanding of the constructed model.*

We have added the Moho in the profiles in Fig. 8.

*m5: page 5 line 21: how does the uncertainty in the empirical relationships of Christensen and Mooney 1995 affect your results?*

These uncertainties might be significant and correspond up to approximately 30 mGal in terms of the gravity field. Therefore, making the inversion we allow for additional corrections of the crustal densities. This is clarified in the revised manuscript.

*m6: textual detail: page 13 line 18 and 21 "as mentioned above" was used twice. Also, this is a bit ambiguous, is it about the sentence, section, or whole paper above this line.*

Corrected.

m7: page 18 line 23: "long-wavelength fileld" -> field. Nad what is meant with longwavelength, specify with d/o.

The boundary wavelength corresponds to the maximum resolution of the density model (1x1 degree), therefore it is equal to approximately 180 d/o, which is clarified.

---

## Author Comment (AC3) · 11 May 2018

Dear Prof. Mandea,

We highly appreciate the constructive suggestions of the reviewers and your editorial comments. The manuscript has been revised accordingly. In particular, we strengthen the discussion about the data quality and robustness of the results. We have also improved the quality of the figures following the reviewers' comments. For example, following the advice of Dr. Root, in the revised manuscript we add a zoomed figure, which demonstrates variations of the maximal value of the horizontal gradient of the isostatic anomalies for Sinai and surrounding area together with the seismicity. With this figure, we also show the magnetic anomaly map, which demonstrates interesting

correlation with the gravity field. We believe that the revised manuscript fits the high standards of the journal.

With kind regards

M. Kaban on behalf of the co-authors

―――――――――――――――――――

---

## Editor Comment (EC2) · M. Mandea (Editor) · 28 May 2018

The provided answers indicate that the initial manuscript can be improved; a new version is expected.

———————————————

---

## Author Response (AR2)

Dear Prof. Mandea,

Thank you for your editorial comments. We have corrected the manuscript accordingly.

Below, we provide a point by point response and attach the manuscript with marked changes.

We thank you for the attention that you are giving to our paper.

5    With kind regards

M. Kaban on behalf of the co-authors

**Response to the editorial comments** (the original comments are in italic).

*The authors have made an important effort to improve the initial manuscript and to provide answers to the referee comments. A few minor changes*

10   Thank you for the positive evaluation of our work.

*1. Equation 1, all variables need to be named and explained (ex ro_ref, etc). Units need also to be indicated when missing (h, M, etc). The same for equation 2.*

According to the editor's suggestions we named all variables and indicated all units when necessary.

*2. Conclusions need to be better structured. The 10 points could be rearranged around some main items, as density*
15   *variation, correlation between the variations of the isostatic anomalies and seismicity, and geodynamical interpretation.*

We have rearranged the conclusions along with the main findings.

*3. Caption of Figure 11 - it might be useful to indicate the period over with the earthquakes are shown.*

The source of the earthquakes and the period of their acquisition are specified in the revised manuscript.

[revised manuscript text omitted]